# Predictive Biomarkers of Treatment Response in Major Depressive Disorder

**DOI:** 10.3390/brainsci13111570

**Published:** 2023-11-09

**Authors:** Louise A. Stolz, Jordan N. Kohn, Sydney E. Smith, Lindsay L. Benster, Lawrence G. Appelbaum

**Affiliations:** 1Department of Psychiatry, University of California San Diego, La Jolla, CA 92093, USA; lostolz@health.ucsd.edu (L.A.S.); jokohn@health.ucsd.edu (J.N.K.); llbenster@health.ucsd.edu (L.L.B.); 2Herbert Wertheim School of Public Health and Human Longevity Science, University of California San Diego, La Jolla, CA 92093, USA; 3Department of Cognitive Science, University of California San Diego, La Jolla, CA 92093, USA; s1smith@health.ucsd.edu; 4Department Clinical Psychology, San Diego State University, San Diego, CA 92182, USA

**Keywords:** depression, predictive biomarkers, electroencephalography, transcranial magnetic stimulation

## Abstract

Major depressive disorder (MDD) is a highly prevalent, debilitating disorder with a high rate of treatment resistance. One strategy to improve treatment outcomes is to identify patient-specific, pre-intervention factors that can predict treatment success. Neurophysiological measures such as electroencephalography (EEG), which measures the brain’s electrical activity from sensors on the scalp, offer one promising approach for predicting treatment response for psychiatric illnesses, including MDD. In this study, a secondary data analysis was conducted on the publicly available Two Decades Brainclinics Research Archive for Insights in Neurophysiology (TDBRAIN) database. Logistic regression modeling was used to predict treatment response, defined as at least a 50% improvement on the Beck’s Depression Inventory, in 119 MDD patients receiving repetitive transcranial magnetic stimulation (rTMS). The results show that both age and baseline symptom severity were significant predictors of rTMS treatment response, with older individuals and more severe depression scores associated with decreased odds of a positive treatment response. EEG measures contributed predictive power to these models; however, these improvements in outcome predictability only trended towards statistical significance. These findings provide confirmation of previous demographic and clinical predictors, while pointing to EEG metrics that may provide predictive information in future studies.

## 1. Introduction

Major depressive disorder (MDD) is a mental health condition characterized by persistent feelings of sadness, hopelessness, and a lack of interest or pleasure in daily activities. This widespread condition affects nearly 1 in 20 people globally, making it a leading cause of disability worldwide, and imposing significant economic costs to the world’s healthcare systems [1]. In addition to the widespread prevalence of MDD, relapse rates tend to increase with successive depressive episodes. Relapse rates tend to rise from around 60% after the first episode to 70% after two episodes, and up to 90% in patients who experience more than two episodes [2]. With this chronicity of relapse, many patients develop Treatment-Resistant Depression (TRD), defined as the absence of a clinical response after at least two adequately dosed antidepressant treatments [3]. Given the profound burden of MDD, there is an impetus to identify characteristics of patients, including demographics, clinical symptomology, and neurophysiological measurements, to better understand its underlying etiology and to individualize treatment modalities. 

A prevalent theory of depression postulates that depressive symptoms stem from disrupted activity and connectivity in brain regions involved in mood and cognition [4]. This theory proposes that individuals with MDD have an imbalance in activity in the dorsolateral prefrontal cortex (DLPFC), leading to the dysregulation of activity downstream in the limbic system. Functional Magnetic Resonance Imaging (fMRI) studies have found that patients with MDD tend to have a hypoactive left DLPFC and/or hyperactive right DLPFC [5,6,7], which subsequently leads to the dysregulation of connectivity and control of the limbic system. Similarly, one study using single-pulse Transcranial Magnetic Stimulation (TMS) has demonstrated this frontal imbalance in excitation and inhibition through concurrent measurement of electroencephalography (EEG) [8]. Clinically, repetitive Transcranial Magnetic Stimulation (rTMS) attempts to resolve this imbalance by noninvasively stimulating affected brain regions with magnetic pulses to normalize neuronal activity. Based on this pathophysiology, to reduce the hyperactivity of the right hemisphere, a standard approach is to administer 1 Hz inhibitory rTMS over the right DLPFC (i.e., low frequency right, LFR). Conversely, to increase activity in the hypoactive left hemisphere, a second common approach is to administer 10 Hz excitatory rTMS to the left DLPFC (i.e., high-frequency left, HFL) [9]. Based on the neuropathophysiology of MDD and these hypothesized mechanisms of action of rTMS, it is possible that pre-treatment EEG measures of asymmetry in fronto–cortical excitation may be useful biomarkers of treatment responses to rTMS. 

Historically, many studies have analyzed the contribution of clinical and demographic variables such as age, gender, pre-treatment symptom severity, and duration of illness to predict treatment response [10,11,12]. While findings have been largely inconsistent, in general, age, symptom severity, and illness duration are negatively associated with positive treatment response. No association has been confirmed between gender and response. Clinical factors have demonstrated stronger predictive capabilities for treatment outcome than demographic factors. Recently, there has been a growing movement to also utilize neurophysiological biomarkers for this purpose [13,14,15,16]. Biomarkers are objective features of a physiological state that are extracted from biological, often brain-derived, data [17]. Predictive biomarkers can prospectively differentiate individuals who will respond positively to treatment from those who do not, thus saving both patients and clinicians valuable time and resources. This paper seeks to address the lack of reliable neurophysiological predictors of treatment response by investigating the contributions of demographic, clinical, and neurophysiological measures in predicting the treatment response of patients with MDD. 

To address this gap, this paper will focus on neurophysiological biomarkers derived from EEG recordings of brain electrical activity. Most EEG studies to date have examined oscillatory power across frequency bands with power in the alpha (8–13 Hz) and theta (4–8 Hz) bands among the leading candidates for EEG biomarkers of depression [18,19]. Despite this, there are conflicting findings [20,21], potentially due to the highly variable methods used to calculate band power [22]. With regard to alpha, a prominent hypothesis is that decreased alpha power is associated with increased default mode network activity and lower odds of responding to treatment [23]. While such alpha effects have been reported in treatment-resistant depression, many antidepressants have also been shown to decrease alpha power [23], creating conflict between MDD pathophysiology and the therapeutic effects of antidepressant treatment. The findings surrounding theta power are also equivocal. While several groups have reported that higher rostral anterior cingulate cortex theta activity at baseline leads to greater improvement in depressive symptoms after both neuromodulatory and pharmacological interventions [24,25], other groups report that theta power is elevated in more severe depression, corresponding to increased treatment resistance [26,27]. However, a recent review of fMRI and EEG biomarkers of depression in response to brain stimulation therapy by Klooster et al. [28] ultimately found that task-based frontal–midline theta activity and individual alpha frequency were the most robust EEG biomarkers. As such, there remains a high degree of uncertainty surrounding the viability of using EEG biomarkers to predict MDD treatment outcomes. 

To overcome some of these issues of poor replicability, the TDBRAIN database [29], which contains pre-treatment EEG recordings from over 100 MDD patients, was analyzed. To improve the reliability of the methodology, the publicly available and validated spectral parameterization method (FOOOF toolbox, v.1.1.0 in Python, v.3.8) was used to extract neural responses from the EEG. This method parameterizes the neural power spectrum to disambiguate the contributions of periodic and aperiodic activity to spectral power. To elaborate, the EEG spectrum is composed of both oscillatory, periodic signals and non-oscillatory, aperiodic signals. Periodic signals are typically analyzed for their relation to various cognitive and physiological processes and appear as visible “bumps” in the power spectrum [30,31]. Alternatively, non-oscillatory, aperiodic activity, which appears as a “1/f-like” decrease in power with increasing frequency, is hypothesized to relate to the balance of excitation and inhibition in the brain [32,33]. Disambiguating the contributions of periodic and aperiodic signals to the power spectrum is important for two reasons: first, when determining the power of an oscillation, changes in aperiodic activity can contribute to apparent changes in spectral power, even when power in that specific frequency band has not changed. To appropriately account for oscillatory power, it must be quantified as power-adjusted to account for the aperiodic component. Second, although previously considered to be biological noise, an increasing number of studies now support the functional relevance of the aperiodic signal in information processing, particularly the balance of excitation and inhibition in the underlying cortex [32,34,35,36]. As such, the spectral parameterization method is employed to assess the characteristics of pre-treatment, resting-state EEG that may predict treatment response in MDD patients undergoing rTMS therapy. This is compared to traditional EEG-band power analyses that do not account for aperiodic activity but may be more directly relatable to findings in the past literature.

This paper contains two major analyses of the TDBRAIN dataset. The first seeks to evaluate a mechanistically driven hypothesis regarding frontal cortical asymmetries in predicting treatment response to HFL (10 Hz) or LFR (1 Hz) rTMS treatments that are believed to act on this imbalance. This analysis will focus on the asymmetry in the aperiodic exponent as an index of the balance of excitation to inhibition (E:I). This investigation will further analyze whether this asymmetry differs between MDD and healthy controls, and whether the concordance between pre-treatment frontal asymmetry and rTMS treatment type is related to differential odds of treatment response in MDD patients. It was hypothesized that patients with greater imbalance in the frontal aperiodic exponent would exhibit worse symptoms and that this asymmetry would predict greater or lesser treatment response. Moreover, patients with hypoactivity in the left DLPFC, as shown by a negative exponent asymmetry, will respond better to HFL treatment, and those with hyperactivity in the right DLPFC will respond to LFR stimulation. The second analysis is an exploratory, generalized, whole-brain evaluation of frequency bands to determine if separation of periodic from aperiodic signals leads to the better prediction of treatment response than traditional band power analyses that do not make this separation. It was hypothesized that lower baseline theta power would predict improved outcomes in treatment response since reduced theta could result from an increased aperiodic exponent which indicates a better balance in E:I. Together, these analyses aim to identify neural markers of treatment response to rTMS therapy in individuals with MDD.

## 2. Materials and Methods

### 2.1. Resting-State EEG Acquisition 

Data for this study originate from the TDBRAIN dataset provided by van Dijk and colleagues [29], which was shared with the scientific community to improve replicability of EEG biomarker identification in mental illness. This dataset consists of raw, resting-state EEG for over 1200 individuals with a variety of clinical diagnoses. Details on the contents and data collection methods of the database can be found in van Dijk et al., 2022 [29]. Briefly, two-minutes of eyes-open and eyes-closed data were collected with 26-channel EEG along with other demographic, lifestyle, and psychological variables, including age, dietary habits, working memory performance, and psychological scales. 

### 2.2. Participants and rTMS Therapy

For the MDD group, patients took part in a combined rTMS and psychotherapy clinical trial [37] where clinical data and treatment responses, as measured through the psychometrically validated Dutch language version of Beck’s Depression Inventory (BDI-II-NL, referred to as “BDI” throughout), were recorded [38]. Only individuals with a clinically adjudicated diagnosis of MDD based on the DSM-5 who completed rTMS treatment were included in analyses. Patients were treated with psychotherapy and at least ten sessions of rTMS, either applied over the left DLPFC at 10 Hz, the right DLPFC at 1 Hz, or both sequentially. Psychotherapy involved 45 min of cognitive behavioral therapy, but specific approaches were tailored to the needs of the patients [37]. The BDI was administered before the first session, then every fifth session until the end of the treatment, and 6 months after the final session. Responders were classified as those whose BDI score improved by 50% or more from baseline to the final session.

### 2.3. EEG Preprocessing 

Prior to analysis, all data were cleaned using the following steps in MNE Python, v.1.2.0 [39]. First, non-brain EEG channels VPVA, HPHL, HNHR, Erbs, OrbOcc, Mass, and VNVB were excluded. Bad channels were marked for interpolation through visual inspection of the raw data. Channels were considered for interpolation if they had substantial drift or consistent periods of high-frequency activity (greater than 2 s). Next, the EEG was re-referenced to the average of all channels, filtered between 0.5 and 80 Hz, notch-filtered at 50 Hz to remove line noise, and run through the infomax method of ICA with 25 components. Eye movement components were rejected based on a 1.96 z-score threshold using the FP1 and FP2 channels. The MNE-ICALabel function [40] was used to automatically detect and reject noise components including eye, muscle, line and channel noise and ECG, which left approximately 80% of the ICs. Any component labeled as “brain” or “other” was kept. Finally, the cleaned data were inspected visually for any remaining noise. If there were still instances of noise—substantial drift or consistent periods of high frequency activity—the remaining “brain” and “other” components were manually inspected and rejected if deemed to have confounding noise. Of the 121 initial MDD individuals, 20 needed extra cleaning beyond the automatic procedures, and one was rejected entirely for poor data quality, leading to 120 individuals who were subsequently processed through the spectral parameterization (FOOOF) pipeline described below. Only the eyes-opened EEG resting data were analyzed to avoid exaggerated alpha power seen during eyes-closed EEG [41].

### 2.4. EEG Analysis

The purpose of the FOOOF algorithm is to parameterize power spectra to characterize signals of interest and overcome limitations of traditional narrowband analyses [33]. The FOOOF algorithm extracts both aperiodic (exponent and offset) and periodic components of the power spectra (center of frequency, bandwidth, power above the aperiodic exponent slope, and total band power). These parameters were extracted for the delta (1–4 Hz), theta (4–7 Hz), alpha (7–12 Hz), beta (12–30 Hz), and gamma (30–45 Hz) bands. EEG data were converted from time-series data into frequency representations using Welch’s method. The FOOOFGroup method [33] was then applied to fit the power spectra, aperiodic and periodic components were independently parameterized, and the model fit was evaluated using the r-squared value. Of the 120 patients run through the FOOOF model, one individual was excluded from further analyses due to poor model fit because over 50% of channels had poor fits in which the r-squared was under 80% for this person. More details on the model can be found in Donoghue et al., 2020 [33], and the code used for this analysis can be found at https://github.com/fooof-tools/fooof (accessed on 20 November 2022).

Next, missing EEG data, which corresponded to a lack of a significant “bump” or peak above the aperiodic component, were accounted for by either excluding them entirely or imputing missing values. All delta frequency variables were excluded from subsequent analysis due to >70% missingness. Missingness was present in 11.8% of the theta samples and 3.4% of the gamma samples and assumed to be missing at random. Therefore, these frequency bins were imputed using single imputation by chained equations, applying the random forest method. The output of FOOOF includes both periodic and aperiodic EEG parameters for all channels and all frequency bands. The resulting data table was averaged, per participant, across all EEG channels for subsequent whole brain analyses and averaged across over channels FC3, F3, and F7 and F4, FC4, and F8 to create left and right DLPFC ROIs for subsequent lateralization analyses to explore exponent asymmetry. Asymmetry was defined as the arithmetic difference between average left and average right ROIs (e.g., right—left). 

### 2.5. Predictive Modeling

In the current study, two sets of analyses were conducted: a lateralized ROI analysis and a whole-head analysis. In both cases, models including EEG parameters were compared against a base model with clinical and demographic variables for their prediction of whether patients responded to rTMS. The base logistic regression model Equation (1) included non-EEG factors including gender, and pre-treatment BDI scores, which were previously shown to associate with rTMS treatment response [42]. 

The lateralized ROI model sought to evaluate a mechanistic hypothesis about neuroplasticity induced by HFL (n = 43) versus LFR (n = 70) TMS, and, specifically, if cortical asymmetry measured at baseline interacted with treatment in a lateralized manner. For this analysis, cortical asymmetry was calculated as the aperiodic exponent from the left ROI subtracted from the right ROI. For the MDD patients, only participants from the HFL and LFR treatment groups were included (five participants with MDD received bilateral rTMS and were excluded from this analysis). As listed in Equation (2), the parameters of this model included the non-EEG parameters from the base model, as well as exponent asymmetry, right ROI exponent as a reference for asymmetry directionality, and the interaction between exponent asymmetry and the treatment protocol (HFL or LFR) each participant received. For this model, the outcome variable was treatment response. Additional regression analyses were run comparing the 36 age-matched healthy controls and the MDD group to determine whether exponent asymmetry (outcome variable) was associated with disease state Equation (3) and whether exponent asymmetry was associated with baseline symptom severity among MDD patients Equation (4). Both included the additional parameter of the right-sided exponent to control for directionality of asymmetry.

The whole-head analysis consisted of two models, each of which included all 119 patients, irrespective of which treatment they received, to evaluate whether broadband characteristics of baseline EEG predicted treatment response. In the first of these models, referred to as the aperiodic model Equation (5), the non-EEG parameters were included along with the aperiodic adjusted power from theta, alpha, beta, and gamma bands, and the exponent calculated according to the FOOOF procedures. In the second whole-head model, referred to as the band power model Equation (6), the base model parameters were included along with the total band powers for each frequency band. All data were analyzed in Python and R (v.4.2.0) and model equations are outlined below:

Base model:
Response ~ age + pre-BDI + gender(1)

Lateralized ROI model:
Response ~ age + pre-BDI + gender + exponent_asymmetry + right_exponent + exponent_asymmetry:rTMS_protocol(2)

MDD vs. healthy:
exponent_asymmetry ~ age + gender + right_exponent + diagnosis(3)

Exponent asymmetry and symptom severity:
exponent_asymmetry ~ age + gender + right_exponent + pre-BDI(4)

Whole-head “aperiodic model”:
Response ~ age + pre-BDI + gender + theta_power + alpha_power +  beta_power + gamma_power + exponent(5)

Whole-head “band power model”: Response ~ age + pre-BDI + gender + theta_band-power + alpha_band-power + beta_band-power + gamma_band-power(6)

### 2.6. Statistical Analyses

Demographic and clinical factors were compared between MDD patients who did not respond and those who did respond to rTMS treatment (defined as ≥50% reduction from pre-treatment BDI scores) using t-tests (for continuous variables) and chi-square tests (for categorical variables). Prior to implementation of logistic regression models, correlation analyses were conducted to evaluate potential multicollinearity of predictors. Only theta and alpha band powers were intercorrelated above 0.40 (*R*^2^ = 0.65). To quantify multicollinearity, variance inflation factors (VIFs) for each parameter were evaluated in all models. In the band power model, VIFs were ≤4, and in the aperiodic model they were ≤2, indicating that multicollinearity was of limited concern [43]. 

Logistic regression models were implemented with rTMS treatment response status (i.e., response versus non-response) as the outcome variable. Model predictors were standardized (mean = 0, SD = 1) and logistic regression models with robust sandwich variance-covariance estimation were fit to the original data using the lrm function within the Hmisc package in *R*. The predictive performance of each model in discriminating individual treatment responses was evaluated against the original data and receiver operating characteristic (ROC) area under the curves (AUCs) were computed. AUC reflects the model accuracy, and as a general rule, 0.70 represents acceptable accuracy, 0.80 reflects good accuracy, and 0.90 is excellent accuracy [44]. Due to the small sample size, the data were not split into training and test sets for fitting and validation, respectively. Instead, all models were fit on full dataset, with correction for model overfitting carried out using Harrell’s optimism correction procedure [45]. Briefly, the original data were vector bootstrapped (*B* = 1000) through stratified random sampling with replacement so that each bootstrapped sample maintained the same ratio of rTMS treatment responders to non-responders as the observed data. Imputation (for whole-head models only) and predictor standardization were performed on each bootstrap sample, followed by Generalized Linear Model (GLM) prediction model construction. Predictive performance was estimated for each *B* prediction model against the bootstrap sample (θ*_1, boot_*, θ*_2, boot_*, … θ*_1000, boot_*) and then used to evaluate performance for the original data (θ*_1, orig_*, θ*_2, orig_*,… θ*_1000, orig_*). Optimism in AUC was estimated through Equation (7) and subtracted from the apparent performance of the original models.
(7)$$\Lambda =\frac{1}{B}\sum_{b=1}^{B}\left(\theta_{b,boot}-\theta_{b,orig}\right)$$

Formal tests for ROC curve differences were carried out using DeLong’s method with the roc.test function in the pROC package in R. Nagelkerke’s R^2^ was also computed [46], where 0–20% indicates a weak relationship between predictors and outcome, 20–40% indicates a moderate relationship, and >40% indicates a strong relationship. Confidence intervals (CI) of 95% are presented for parameter estimates throughout logistic regression models, as well as bias-corrected accelerated (BC_a_) CIs from bootstrapped samples [47]. Post hoc testing of the lateralized ROI model Equation (2) was performed by computing estimated marginal means at −2, −1, 0, 1, and 2 SDs from the standardized mean exponent asymmetry score for the sample for HFL and LFR rTMS treatment groups and applying false discovery rate (FDR) correction to the set of *p*-values (n = 5) from these post hoc contrasts (e.g., response probability for participants with −2-SD asymmetry who received HFL versus LFR, etc.).

## 3. Results

### 3.1. Group Differences

As a preliminary analysis, the demographics of MDD patients and healthy controls are listed in Table 1 below. Of the 119 MDD patients (mean age: 43.1 years old, SD: 13.3, range: 19–78) with available pre-treatment EEG data, 66% were categorized as treatment responders (n = 78) and 50% were female (n = 60; Table 1). There were no significant differences between responders and non-responders with respect to age, gender, pre-treatment BDI scores, or the proportion who received HFL, LFR, or bilateral rTMS treatment. 

### 3.2. Lateralized ROI Model Results

In the first analysis, to evaluate the role of lateralized exponent asymmetries, logistic regression analyses were performed on the 113 patients who received either HFL or LFR rTMS. Participants who completed bilateral rTMS (n = 5) were excluded, as the purpose of this analysis was to examine the lateralized effect of treatment, which would not be possible in subjects who received stimulation to both brain hemispheres. As a first step, to determine whether the hemispheric asymmetry in pre-treatment exponent values was related to MDD symptom severity Equation (4) or differentiated MDD patients from healthy controls Equation (3), linear regression models were performed with these factors included. The results showed that when adjusted for age, gender, and right ROI exponent, there was no association between pre-treatment BDI and exponent asymmetry in MDD patients (β = −0.04, 95% CI: −0.23–−0.15, *p* = 0.70). Similarly, there were no differences in exponent asymmetry between MDD patients and healthy controls (β = −0.03, 95% CI: −0.41–0.36, *p* = 0.89). 

As a second step, logistic regression models were implemented to determine whether baseline hemispheric asymmetry in exponent power predicted treatment response. In an initial “base model” (Figure 1A), which contained only clinical and demographic factors, these variables explained approximately 9.9% of the variance in treatment response. Within this model, pre-treatment BDI scores were the only significant predictor, wherein for every 1-SD (9.8 point) increase in pre-treatment BDI scores, there were 39% lower odds of a positive treatment response (OR = 0.61, 95% CI_BCa_ = 0.35–1.02). Age and gender were not associated with differential treatment responses, though older patients tended to have lower odds of a positive treatment response.

Next, to test whether hemispheric asymmetries in pre-treatment EEG exponent power were associated with different responses to the two lateralized rTMS protocols, a second, “biomarker model” was tested (Figure 1). This model included age, gender, and pre-treatment BDI from the base model, along with the right ROI exponent value, the exponent asymmetry, the rTMS protocol, the interaction between the asymmetry and protocol. Overall, these factors explained 19.4% of the variance in treatment response (R^2^ = 0.194), which was 9.5% more than the base model alone. In this model, age and pre-treatment BDI were significant predictors; however, the overall fit of the biomarker model only marginally improved relative to the base model (likelihood ratio test: $X_{4}^{2}$ = 8.64, *p* = 0.07). Within this trending effect, there was evidence to suggest that the probability of treatment response varied as a function of exponent asymmetry for the two rTMS protocols. Figure 1b shows the interaction between treatment protocol, exponent asymmetry, and treatment response. 

To elaborate more on Figure 1b, for patients whose exponent values were greater in the left hemisphere than the right (negative asymmetry), treatment responses to HFL and LFR differed significantly. The negative asymmetry of 1- and 2-SD below the sample mean would be expected to have 15% (95% CI: 3–26%, *p_FDR_* = 0.03) and 21% (12–30%, *p_FDR_* < 0.001) poorer treatment responses, respectively, for patients who received HFL compared to LFR. 

### 3.3. Whole-Head Model Results

In the second analysis of this paper, the whole-head model (including all 119 patients) sought to determine whether the separation of periodic from aperiodic EEG signals led to a better prediction of treatment response than traditional band power calculations. A primary finding was that the initial base model, including only demographic and clinical features, indicated that both older age and higher pre-treatment BDI scores were associated with significantly lower odds of a positive treatment response to rTMS (Figure 2A). Together, these variables explained approximately 11.9% of the variance in treatment response and, specifically, a 1-SD increase in pre-treatment BDI score (+10.2 points) and age (+13.3 years) corresponded to 42% lower odds of responding to treatment. 

Subsequently, the addition of band power EEG features (Figure 2B) and aperiodic EEG features (Figure 2c) produced models that explained an additional 10.1% (R^2^ = 0.22) and 13.6% (R^2^ = 0.255) of the variance in rTMS treatment response, respectively. Both models yielded improved fit to the observed data relative to the base model (likelihood ratio test: $X_{4}^{2}$ = 9.90, *p* = 0.04; $X_{5}^{2}$ = 13.6, *p* = 0.02), with the aperiodic model producing a somewhat better fit than the band power model ($X_{1}^{2}$ = 3.67, *p* = 0.06). As shown graphically in the bottom of each panel of Figure 2, the base model was able to discriminate between responders and non-responders with fair accuracy (ROC AUC = 67.1%), while the band power model (ROC AUC = 74.4%) and aperiodic model (ROC AUC = 74.5%) improved these predictions by about 7% each. Statistically, these improvements were marginal for both the band power model (Delong’s test: *Z* = 1.76, 95% CI: −0.01, 0.15, *p* = 0.078) and the aperiodic model (Z = 1.76, 95% CI: −0.01, 0.16, *p* = 0.079). To adjust for bias due to model overfitting, optimism correction was performed on 1000 bootstrapped samples for the base model and both EEG models, which indicated that the prediction for both the band power and aperiodic EEG models was fair (ROC AUC = 68.0% and 67.4%, respectively). Additional details of the model performance are shown in Table 2. 

Despite the relatively small discriminatory improvement in the EEG models relative to the base model, individual pre-treatment EEG features were predictive of treatment response; however, since the overall model was non-significant, these findings should be carefully interpreted. In the band power model, higher pre-treatment gamma power (OR = 1.95, 95% CI = 1.09–3.49; 95% CI_BCa_ = 1.34–4.97) and lower beta power (OR = 0.46, 95% CI = 0.22–0.97; 95% CI_BCa_ = 0.21–1.02) were associated with greater odds of treatment response to rTMS. By contrast, in the aperiodic model, neither gamma nor beta aperiodic-adjusted power (i.e., the narrowband response once the aperiodic exponent was separated) was associated with differential odds of treatment response. However, lower theta amplitude was associated with greater odds of a positive response to rTMS (OR = 0.47, 95% CI = 0.26–0.86; 95% CI_BCa_ = 0.29–1.01).

## 4. Discussion

Overall, this secondary analysis of the TDBRAIN dataset identified demographic and clinical characteristics that delineated rTMS treatment responders from non-responders. Older age and greater baseline depressive symptom severity were significant predictors of poorer treatment responses, while EEG only marginally improved predictive performance. Furthermore, differences in frontal exponent asymmetry were not associated with symptom severity or with MDD diagnosis. The following discussion describes the model findings in greater detail, addresses the strengths and weaknesses in this approach, and makes suggestions for future studies that may add new knowledge to the emerging field of biomarker development. 

### 4.1. Age and Baseline BDI as Predictors of Treatment Response

These findings align with previous studies indicating that age and baseline symptom severity are negatively associated with treatment outcome [48,49,50,51]. Results from the whole-head model revealed that for each additional 13 years of age, there was a 42% reduction in the odds that a patient will respond to rTMS. Several possible explanations for this relationship exist. One hypothesis is that neuroplasticity in the recovery of dysfunctional network connectivity is reduced with age, especially in the prefrontal cortex [52,53,54,55], the target brain region for rTMS therapy. Such a reduction in neuroplasticity would decrease the efficacy of TMS in potentiating or de-potentiating target neurons. Moreover, age is also associated with neuroinflammation, amyloid accumulation, mitochondrial dysfunction, and physical comorbidities which can contribute to overall neural dysfunction, rendering treatment less effective [56]. Another possibility is that in older patients, cerebral ventricles tend to enlarge and cortical volume decreases, leading to a larger distance between the TMS coil and fronto–cortical targets, and thereby a reduction in the effective magnetic field strength [57,58]. While stimulation parameters are typically adjusted to the patient’s motor threshold, which should account for some of the differences in coil-to-target distance, this adjustment may insufficiently account for the larger relative differential in depth between the motor and frontal cortices typically observed in older adults. As such, the activation elicited by the coil may not reach deeper regions linked with the DLPFC, leading to lower therapeutic benefits for older adults. However, these hypotheses are speculative and should be lightly interpreted.

Another result observed here was that a 10-point increase in pre-treatment BDI scores was associated with 42% lower odds of a positive treatment response. Higher baseline symptom severity associated with reduced clinical response has been attributed to treatment-resistant symptoms such as anhedonia [59,60,61], which are particularly resistant to therapy. Future studies may wish to analyze domain-specific symptomatology to ascertain whether such symptoms improve the pre-treatment prediction of TMS response probabilities and the extent to which treatment responses themselves are domain-specific. It is possible that patients with these persistent symptoms have similar underlying neuropathology and would most likely not respond to other neurostimulation therapies. 

### 4.2. Lateralized ROI Model Findings

For this analysis, it was hypothesized that patients with greater imbalance in the frontal aperiodic exponent would exhibit worse symptoms and that this asymmetry would predict a greater or lesser treatment response. Analyses testing the association between symptom severity and exponent asymmetry did not reveal any relationship, nor did it identify differences between MDD patients and healthy controls. These null results may be due to the small sample size of the healthy control group or because disrupted information processing is not strongly associated with MDD. Future studies should analyze whether changes in the exponent after rTMS are associated with treatment response, as this may demonstrate whether treatment can restore excitatory–inhibitory (E:I) balance. 

The lateralized model sought to test whether the interaction between treatment protocol and exponent asymmetry-predicted treatment response based on the hypothesis that patients with a higher imbalance in the exponent would be more amenable to the lateralized treatments that are designed to restore E:I balance. Overall, patients with relatively hypoactive left or hyperactive right frontal activity who received HFL stimulation did not respond as well to treatment; however, LFR was relatively more effective in those with hyperactive right frontal activity. While these results must be interpreted with extreme caution due to the limited statistical significance of the model and the lack of post-treatment EEG measures that could illustrate within-subject changes in asymmetry, the observed pattern can be interpreted as one of two possibilities. First, that activity in the left hemisphere might be characterized by a relatively greater inhibitory drive, a potential indicator of hypoactivity, or second, the right hemisphere might be characterized by greater excitatory drive, a potential indicator of hyperactivity [6]. Seeing that LFR, which aims to quiet a hyperactive right hemisphere, was more effective for these patients, we can speculate that targeting patients with right hemisphere hyperactivity using LFR stimulation is a potentially optimal treatment strategy, as opposed to HFL. This finding contrasts with expectations, as one would expect that patients with a hypoactive left frontal cortex would respond better to HFL stimulation, which seeks to address the imbalance. Notwithstanding, this finding warrants future investigation into the therapeutic efficacy of LFR as a function of hemispheric asymmetry in aperiodic activity. 

Finally, the base and EEG biomarker models were compared for their ability to predict treatment outcome. Overall, findings were not significant, but adding exponent asymmetry as a predictor to the model yielded an improvement (~9.5%) in predicting treatment response over the base model. 

### 4.3. Whole-Head Model Findings

The second half of this paper sought to evaluate band power EEG prediction while also extending and comparing this analysis to EEG processed through the FOOOF algorithm [33] to separate periodic from aperiodic signals. To date, only one other study has separated these powers to analyze aperiodic power in MDD compared to healthy brains; however, this was done in silico [62], making the current analysis a novel application towards predicting clinical treatment response. 

In this analysis, two EEG models were compared to a base demographic model that included age, gender, and pre-treatment BDI scores. Similar to the lateralized ROI analysis, the base model here significantly predicted treatment response, explaining nearly 12% of the overall variance, with both older age and higher pre-treatment BDI scores associated with significantly lower odds of a positive treatment response to rTMS. Also, similar to the lateralized ROI model, the addition of EEG features in both the band power and aperiodic models led to an improved accuracy in predicting treatment response by approximately 8% compared to the base model, but these differences only trended towards significance, with both models performing similarly to one another (delta AUC: *p* = 0.96). 

While these results did not meet the significance criteria, it is interesting to note that the pattern of features did differ between the two models. In the band power model, higher gamma and lower beta power were associated with improved odds of treatment response, whereas in the aperiodic model, lower theta aperiodic-adjusted power was associated with improved response odds. The capacity to harmonize these models is limited, as band power and aperiodic-adjusted power are different metrics that quantify different phenomena: band power is an absolute measure of all power in the frequency band, while periodic power is adjusted for the contribution of aperiodic activity. However, higher gamma band power and lower beta band power could indicate that treatment responders have flatter power spectra at baseline. Compared to non-responders, this would appear like a see-saw rotation in the spectrum with a fulcrum at a frequency (~30 Hz) between the beta and gamma bands, as reported by Donoghue and colleagues [33]. While it is not possible to conclude that these patterns are associated with treatment response, or to opine on the underlying mechanisms, the presence of these differing patterns suggests that they may be candidates for further investigation. 

### 4.4. Limitations and Further Considerations

Although this project utilized a large sample size and novel analysis methods, there were several limitations which should be addressed in future studies. First, only two minutes of resting-state data were available, which may have limited the signal-to-noise ratio of the spectra analyses, and also resulted in the inability to extract reliable slow-wave delta peaks. As such, delta was excluded as a predictor due to high missingness (>70%). Second, although the original TDBRAIN dataset included EEG from 426 MDD patients, only a subset (n = 121) received a clinically adjudicated diagnosis which reduced the sample size of this study. Furthermore, while the TDBRAIN repository includes a hold-out sample for model validation, this group consists of multiple psychiatric conditions for whom pre-treatment BDI scores are not available. Because these are critical inputs for the models in this study, it was not possible to cross-validate the findings in the available holdout sample. Finally, model performance would most likely be improved with more demographic information, which is typically available to clinicians, such as the number of past failed treatments, duration of current illness, number of TMS sessions, and concomitant medication use. The addition of these variables could potentially address the gap in predictive performance between the base and EEG models. 

Future studies may wish to build on the current analyses to address several questions. First, this patient population received psychotherapy in conjunction with rTMS, which may impact treatment response rates. Future studies could evaluate treatment only involving rTMS, pharmacotherapy and rTMS, or electroconvulsive therapy. These analyses could further reveal similarities in neurophysiology between individuals who respond positively to different neurostimulation therapies. Finally, an area of important concern is the change in neurophysiology from pre- to post-treatment. Future studies should aim to look specifically at changes in EEG before and after therapy to better understand how neurostimulation affects neurophysiology. 

## 5. Conclusions

Overall, these findings support past observations that pre-treatment symptom severity and patient age both negatively impact rTMS treatment outcomes. The presence of trending, but non-significant, results from the EEG models suggests that while these measures of neurophysiology could have a relationship with treatment outcomes with the available data in this study, they are not sufficient to make strong claims. Future research with a larger sample and longer recording durations may lead to better predictions. 

## Figures and Tables

**Figure 1 brainsci-13-01570-f001:**
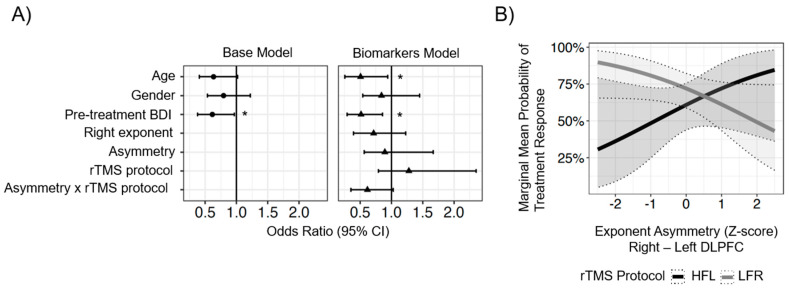
(**A**) Odds ratios with 95% bias-accelerated bootstrapped CI for the base model (circles) and the biomarker model (triangles). Base model includes age, gender, and pre-treatment BDI scores. Biomarker model includes base model and exponent power in the right ROI, exponent asymmetry calculated as right ROI minus left ROI, the rTMS protocol, and the interaction between rTMS protocol and exponent asymmetry. * *p* < 0.05, FDR-corrected. (**B**) Marginal mean probabilities of treatment response as a function of exponent asymmetry and rTMS protocol. HFL = high-frequency left. LFR = low-frequency right.

**Figure 2 brainsci-13-01570-f002:**
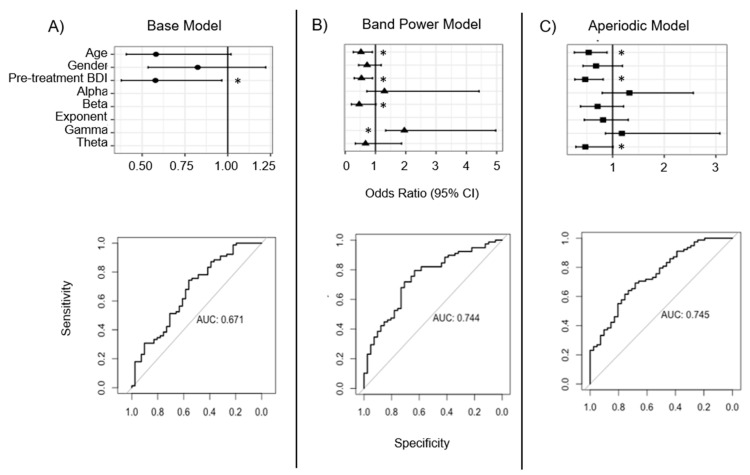
Odds ratios (top) and receiver operating characteristic curves (bottom) from logistic regression for (**A**) base (circles), (**B**) band power (triangles), and (**C**) aperiodic EEG models (squares). Odds ratios shown with 95% bias-corrected accelerated confidence intervals from 1000 bootstrapped samples. * *p* < 0.05, uncorrected. Theta, gamma, beta, and alpha refer to mean band power (band power model) or amplitude values (aperiodic model) across the whole head at the pre-rTMS assessment.

**Table 1 brainsci-13-01570-t001:** Participant characteristics: Descriptive information and statistical comparisons. T-tests and chi-square tests were performed on the MDD group only and test statistics compared responders and non-responders. HFL = high-frequency (10 Hz) left TMS. LFR = low-frequency (1 Hz) right TMS. BI = bilateral (HFL + LFR) TMS. Numbers in parentheses represent standard error.

	MDD Responder	MDD Non-Responder	Test Statistics	Healthy Controls
Number of participants	78	41		36
Average age (years)	41.4 (1.4)	46.3 (2.3)	t_72_ = 1.85, *p* = 0.07	32.2 (2.3)
Number male	37	22	$X_{1}^{2}$ = 0.20, *p* = 0.65	0.42
Pre-treatment BDI	30.1 (1.0)	33.6 (1.8)	t_66_ = 1.67, *p* = 0.10	N/A
Post-treatment BDI	7.2 (0.6)	28.4 (1.7)	t_51_ = 11.4, *p* < 0.001	N/A
rTMS protocol (% HFL/LFR/Bi)	36/64/3	41/59/2	$X_{1}^{2}$ = 0.07, *p* = 0.79	N/A

**Table 2 brainsci-13-01570-t002:** Performance metrics showing accuracy and model fit. AUC is shown with associated BCa CI.

	Base Model	Band Power Model	Aperiodic Model
ROC AUC	67.1% (58.5, 78.9)	74.4% (68.7, 85.5)	74.5% (68.1, 85.4)
Optimism-corrected AUC	63.6% (−3.5%)	68.0% (−6.4%)	67.4% (−7.1%)
Nagelkerke R^2^	11.9%	22.0%	25.5%
Delta-AUC (vs. Base)		Z = 1.77, *p* = 0.078	Z = 1.76, *p* = 0.079
Delta-AUC (vs. BP)			Z = 0.05, *p* = 0.96

## Data Availability

Data were made available through the Brainclinics foundation: https://brainclinics.com/resources/ (accessed on 20 November 2022).

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
