# Peer review of "Predictive Biomarkers of Treatment Response in Major Depressive Disorder"

_brainsci, 2023, doi:10.3390/brainsci13111570_

Round 1

Reviewer 1 Report

Comments and Suggestions for Authors

Thank you for the opportunity to review this work.

However, the manuscript still has the following problems worthy of attention, through the improvement of these problems can better improve the quality of the manuscript.

Abstract

The abstract should summarize key details from all main sections of the paper (intro, methods, results, discussion). Currently, it focuses heavily on the methods and results. Add 1-2 sentences summarizing the research motivation, background, and implications.

Clarify the outcome/dependent variable - what defines treatment response vs non-response? Be specific.

Reduce the level of detail provided on methods and focus on key elements. The abstract should give an overview, not repeat the methods section.

Move non-essential details (e.g. specific p-values) to the main text.

Break up some long sentences for better readability.

Introduction

The introduction covers a broad scope of background information and hypotheses. Consider narrowing the focus to the key concepts most relevant to the current study aims and analyses. Less pertinent details could be condensed or removed.

The description of the study rationale is a bit unclear. The introduction would benefit from more explicitly stating the purpose, novelty, and potential impact of the current study. What gap in the literature is this study addressing?

The hypotheses could be more clearly stated, especially for the second exploratory analysis. Precisely clarify the study predictions.

Review the flow and organization of the background information. Some paragraphs seem out of sequence logically. Ensure the background builds a solid framework leading into the study goals.

Tighten up the writing to be more concise. Some passages contain wordy or repetitive phrasing. Aim for clarity and brevity.

Improve transitions between paragraphs. Consider starting new sections with topic sentences.

Method

The section provides extensive detail but the overall organization could be improved for clarity. Consider structuring into subsections with informative headings such as Participants, EEG Acquisition, EEG Preprocessing, ROIs, Predictive Modeling, etc.

The participant details are currently dispersed throughout. Condense the information about the sample, inclusion/exclusion criteria, diagnoses, treatments, and clinical assessments into a dedicated Participants subsection.

The model descriptions are unclear in places. For each model, restate the main purpose, specific predictors, and outcome variable. The model equations help but verbal explanations are also needed.

Review flow and transitions between paragraphs. Some sections jump between topics without linkage.

Shorten lengthy paragraphs when possible. Break up dense blocks of text for easier reading.

Clarify language when vague (e.g. "substantial drift", "consistent periods"). Quantify if possible.

Results

Clarify the purpose and findings of each analysis upfront before providing statistics. Avoid discussing stats without interpretation.

Explain the rationale and clinical relevance of findings throughout. Don't just state results.

Reduce redundancy. For example, the explanation of rTMS protocols is repeated.

Simplify language and sentence structure for clarity. Some passages contain complex phrasing.

Review use of causal language. Findings are correlational.

The Results cover a lot of important analyses but lack concise explanations for a broad readership. Adding clear signposting and simplifying the language will improve the accessibility of this section. Focus on communicating the key findings and their implications.

Discussion and conclusion

Strengthen links between key results and interpretations. Some conclusions go beyond what the data supports.

Reduce speculation and causal language when describing results, especially null findings. Findings are correlational.

Simplify wording and sentence structure for clarity. Some passages are dense.

Reduce repetition from the Results section. Focus on high-level synthesis and implications.

Author Response

Dear Editor Meng,

We greatly appreciate the time and effort you and the reviewers dedicated to providing insightful and valuable feedback to improve the quality of our manuscript. In response to the comments and suggestions, we have made modifications to address the specific critiques from both reviewers.  As instructed, we have paid careful attention to improving the clarity of our interpretation, removed causal language where results were correlational, and added more specific language to the Methods section. Please see below for responses to each reviewer comment as well as edits to the manuscript which are marked in blue font. We believe that these revisions have greatly strengthened the manuscript and that the reviewers and editorial team will agree the revised version to now be suitable for publication in Brain Sciences. Thank you for your consideration and we look forward to hearing from you with a final decision.

Louise Stolz

Reviewer 1 comments to the author:

Abstract: The abstract should summarize key details from all main sections of the paper (intro, methods, results, discussion). Currently, it focuses heavily on the methods and results. Add 1-2 sentences summarizing the research motivation, background, and implications. Clarify the outcome/dependent variable - what defines treatment response vs non-response? Be specific. Reduce the level of detail provided on methods and focus on key elements. The abstract should give an overview, not repeat the methods section. Move non-essential details (e.g. specific p-values) to the main text. Break up some long sentences for better readability.

Author response: Thank you for your suggestions. As suggested, we have reduced the length of the methods and clarified the implications of our findings. We have also improved the readability by breaking up longer sentences. The new abstract reads:

“Major depressive disorder (MDD) is a highly prevalent, debilitating disorder with a high rate of treatment resistance. One strategy to improve treatment outcomes is to identify patient-specific, pre-intervention factors that can predict treatment success. Neurophysiological measures such as electroencephalography (EEG), which measures the brain’s electrical activity from sensors on the scalp, offer one promising approach for predicting treatment response for psychiatric illnesses, including MDD. In this study, a secondary data analysis was conducted on the publicly available Two Decades-Brainclinics Research Archive for Insights in Neurophysiology (TDBRAIN) database. Logistic regression modeling was used to predict treatment response, defined as at least a 50% improvement on the Beck’s Depression Inventory, in 119 MDD patients receiving repetitive transcranial magnetic stimulation (rTMS). Results show that both age and baseline symptom severity were significant predictors of rTMS treatment response, with older individuals and more severe depression scores associated with decreased odds of a positive treatment response. EEG measures contributed predictive power to these models; however, these improvements in outcome predictability only trended towards statistical significance. These findings provide confirmation of previous demographic and clinical predictors, while pointing to EEG metrics that may provide predictive information in future studies.”

Introduction: The introduction covers a broad scope of background information and hypotheses. Consider narrowing the focus to the key concepts most relevant to the current study aims and analyses. Less pertinent details could be condensed or removed. The description of the study rationale is a bit unclear. The introduction would benefit from more explicitly stating the purpose, novelty, and potential impact of the current study. What gap in the literature is this study addressing? The hypotheses could be more clearly stated, especially for the second exploratory analysis. Precisely clarify the study predictions. Review the flow and organization of the background information. Some paragraphs seem out of sequence logically. Ensure the background builds a solid framework leading into the study goals. Tighten up the writing to be more concise. Some passages contain wordy or repetitive phrasing. Aim for clarity and brevity. Improve transitions between paragraphs. Consider starting new sections with topic sentences.

Author response: Thank you for your detailed feedback. We agree that the introduction could benefit from improvements in clarity and more defined hypotheses. As such, we have incorporated your feedback, restructured the introduction to focus on more pertinent details, and improved the flow of ideas. Please see the changes in the manuscript indicated in blue font.

Methods: The section provides extensive detail but the overall organization could be improved for clarity. Consider structuring into subsections with informative headings such as Participants, EEG Acquisition, EEG Preprocessing, ROIs, Predictive Modeling, etc. The participant details are currently dispersed throughout. Condense the information about the sample, inclusion/exclusion criteria, diagnoses, treatments, and clinical assessments into a dedicated Participants subsection. The model descriptions are unclear in places. For each model, restate the main purpose, specific predictors, and outcome variable. The model equations help but verbal explanations are also needed. Review flow and transitions between paragraphs. Some sections jump between topics without linkage. Shorten lengthy paragraphs when possible. Break up dense blocks of text for easier reading. Clarify language when vague (e.g. "substantial drift", "consistent periods"). Quantify if possible.

Author response: Thank you for the suggestions. We have revised our Methods including greater use of sections headers to communicate each section more clearly per your suggestions. We removed unnecessary descriptions of participants from other sections unless they were specifically related to that analysis (such as which participants were excluded at each stage of processing). As requested, verbal descriptions of the purpose of each model are found in lines 222-248. Finally, we clarified additional variables for each equation, removed vague language when possible, and quantified unclear descriptions.

Results: Clarify the purpose and findings of each analysis upfront before providing statistics. Avoid discussing stats without interpretation. Explain the rationale and clinical relevance of findings throughout. Don't just state results. Reduce redundancy. For example, the explanation of rTMS protocols is repeated. Simplify language and sentence structure for clarity. Some passages contain complex phrasing. Review use of causal language. Findings are correlational. The Results cover a lot of important analyses but lack concise explanations for a broad readership. Adding clear signposting and simplifying the language will improve the accessibility of this section. Focus on communicating the key findings and their implications.

Author response: As suggested, we have improved the clarity of phrasing by reducing its complexity and have added introductory sentences to better contextualize the objectives underlying each analysis. We believe these additions have addressed your comments regarding clarifying the purpose and findings of our results as we had already done this before listing the statistics. We hope that the addition of topic sentences/signposts have made our results clearer. We further confirmed that there is no repetition of the rTMS protocols with each protocol (HFL and LFR) mentioned where appropriate.

Discussion and conclusion: Strengthen links between key results and interpretations. Some conclusions go beyond what the data supports. Reduce speculation and causal language when describing results, especially null findings. Findings are correlational. Simplify wording and sentence structure for clarity. Some passages are dense. Reduce repetition from the Results section. Focus on high-level synthesis and implications.

Author response: We have clarified the interpretations of our results to reduce casual language. In addition, we have incorporated your feedback to reduce repetition from the results as well as simplify sentence structure for clarity. We believe that our interpretation of the results from Section 4.1 is quite clear, yet as noted we have taken effort to reduce the use of casual language and speculation where necessary.

Reviewer 2 Report

Comments and Suggestions for Authors

The manuscript “Predictive biomarkers of treatment response in Major Depressive Disorder” reports an interesting study reports predicting treatment response from baseline demographics, symptom severity, and resting-state EEG features in 119 Major depressive disorder (MDD) patients receiving repetitive transcranial magnetic stimulation (rTMS), conducting secondary data analysis on the publicly available Two Decades-Brainclinics Research Archive for Insights in Neurophysiology (TDBRAIN) database using hierarchical regression modeling.

1.     Line 315-317; Authors should explain the reason behind excluding the bilateral rTMS participants and including either HFL or LFR rTMS data for logistic regression analyses.

2.     Line 357-363; The interpretation of results is not significant as mentions therefore authors must compare these results with reported ones.

3.     Line 413-428; Statements are repeated. Authors should discuss the results comparing with reported literatures with suitable citations.

Author Response

Dear Editor Meng,

We greatly appreciate the time and effort you and the reviewers dedicated to providing insightful and valuable feedback to improve the quality of our manuscript. In response to the comments and suggestions, we have made modifications to address the specific critiques from both reviewers.  As instructed, we have paid careful attention to improving the clarity of our interpretation, removed causal language where results were correlational, and added more specific language to the Methods section. Please see below for responses to each reviewer comment as well as edits to the manuscript which are marked in blue font. We believe that these revisions have greatly strengthened the manuscript and that the reviewers and editorial team will agree the revised version to now be suitable for publication in Brain Sciences. Thank you for your consideration and we look forward to hearing from you with a final decision.

Louise Stolz

Reviewer 2 comments to the author:

Line 315-317: Authors should explain the reason behind excluding the bilateral rTMS participants and including either HFL or LFR rTMS data for logistic regression analyses.

Author response: Thank you for this suggestion. To more clearly delineate our reasoning for leaving out the participants in the bilateral stimulation group, we have edited the text as follows:

“In the first analysis, to evaluate the role of lateralized exponent asymmetries, logistic regression analyses were performed on the 113 patients who received either HFL or LFR rTMS. Participants who completed bilateral rTMS (n=5) were excluded as the purpose of this analysis was to examine the lateralized effect of treatment, which would not be possible in subjects who received stimulation to both brain hemispheres.”

Line 357-363: The interpretation of results is not significant as mentions therefore authors must compare these results with reported ones.

Author response: Thank you for your feedback. We have edited this paragraph as follows to clarify the lack of significance which lends to our cautious interpretation and added references for our interpretations. We have also moved the interpretations in the Results section to the Discussion:

“While these results must be interpreted with extreme caution due to limited statistical significance of the model and the lack of post-treatment EEG measures that could illustrate within-subjects changes in asymmetry, the observed pattern can be interpreted as one of two possibilities. First, that activity in the left hemisphere might be characterized by relatively greater inhibitory drive, a potential indicator of hypoactivity, or second, the right hemisphere might be characterized by greater excitatory drive, a potential indicator of hyperactivity [6]. Seeing that LFR, which aims to quiet a hyperactive right hemisphere, was more effective for these patients, we can speculate that targeting patients with right hemisphere hyperactivity using LFR stimulation is a potentially optimal treatment strategy, as opposed to HFL. This finding contrasts with expectations as one would expect that patients with a hypoactive left frontal cortex would respond better to HFL stimulation that seeks to address the imbalance. Notwithstanding, this finding warrants future investigation into the therapeutic efficacy of LFR as a function of hemispheric asymmetry in aperiodic activity.”

Line 413-428: Statements are repeated. Authors should discuss the results comparing with reported literatures with suitable citations.

Author response: The purpose of this paragraph was to provide a brief summary of our goals and a general overview of our findings. However, we understand that much of this content is repetitive and have followed your suggestions while still providing a short introduction to the Discussion section.  The revised passage reads:

“Overall, this secondary analysis of the TDBRAIN dataset identified demographic and clinical characteristics that delineated rTMS treatment responders from non-responders. Older age and greater baseline depressive symptom severity were significant predictors of poorer treatment responses, while EEG only marginally improved predictive performance. Furthermore, differences in frontal exponent asymmetry were not associated with symptom severity or with MDD diagnosis. The following discussion describes the model findings in greater detail, addresses the strengths and weaknesses in this approach, and makes suggestions for future studies that may add new knowledge to the emerging field of biomarker development.”

Round 2

Reviewer 1 Report

Comments and Suggestions for Authors

NONE

Reviewer 2 Report

Comments and Suggestions for Authors

This revised version of the manuscript responses reviewers' comments satisfactorily, therefore recommended for publication.